# Position: Uncertainty Quantification in LLMs is Just Unsupervised Clustering

**Tiejin Chen, Longchao Da, Xiaoou Liu, Hua Wei** [1]

## Abstract

Uncertainty Quantification (UQ) is widely regarded as the primary safeguard for deploying Large Language Models (LLMs) in high-stakes domains. However, we argue that the field suffers from a category error: mainstream UQ methods for LLMs are just unsupervised clustering algorithms. We demonstrate that most current approaches inherently quantify the internal consistency of the model's generations rather than their external correctness. Consequently, current methods are fundamentally blind to factual reality and fail to detect "confident hallucinations," where models exhibit high confidence in stable but incorrect answers. Therefore, the current UQ methods may create a deceptive sense of safety when deploying the models with uncertainty. In detail, we identify three critical pathologies resulting from this dependence on internal state: a hyperparameter sensitivity crisis that renders deployment unsafe, an internal evaluation cycle that conflates stability with truth, and a fundamental lack of ground truth that forces reliance on unstable proxy metrics to evaluate uncertainty. To resolve this impasse, we advocate for a paradigm shift to UQ and outline a roadmap for the research community to adopt better evaluation metrics and settings, implement mechanism changes for native uncertainty, and anchor verification in objective truth, ensuring that model confidence serves as a reliable proxy for reality.

## 1. Introduction

Large Language Models (LLMs) have demonstrated remarkable ability (Abdin et al., 2024; Touvron et al., 2023; Da et al., 2024b; Luo et al., 2026), yet their deployment in high-stakes domains, such as healthcare and law, remains

---
[1]School of Computing and Augmented Intelligence, Arizona State University. Correspondence to: Hua Wei <hua.wei@asu.edu>.

*Proceedings of the 43rd International Conference on Machine Learning*, Seoul, South Korea. PMLR 306, 2026. Copyright 2026 by the author(s).

challenging due to the issue of hallucinations. To bridge the gap of reliable deployment, the field has rallied around Uncertainty Quantification (UQ) (Liu et al., 2025a; Chen et al., 2026b). By attaching an uncertainty score to a question given the model, we can filter out errors or ensure safety, ranging from entropy-based measures (McCabe et al., 2025; Kuhn et al., 2023) to graph-based methods (Lin et al., 2023; Da et al., 2024a; 2025). In the meantime, researchers find that the inconsistency of generation largely represents the uncertainty of LLMs and becomes mainstream UQ methods for LLMs. However, despite the growth of UQ methods, we face a serious situation: models are becoming confident about their hallucinations, often rendering current uncertainty scores deceptive.

In this paper, we contend that this paradox arises from a fundamental category error of UQ. **We argue that mainstream UQ in LLMs is mechanically isomorphic to an unsupervised clustering problem, which measures internal consistency and fails to serve as a practical safeguard.** Despite their surface differences, we demonstrate that prevailing methods collapse into this single paradigm: Semantic Entropy (Kuhn et al., 2023) functions as explicit clustering by discretizing meanings into "Answer Classes"; graph-based methods (Lin et al., 2023) perform spectral clustering on response similarities; and verbalized method (Kadavath et al., 2022) implicitly clusters internal beliefs. Consequently, these approaches inherit the intrinsic limitation of unsupervised learning: they can only measure the separation of data points, not their semantic correctness. Current UQ methods, therefore, fail to distinguish factual certainty from "confident hallucination" (Simhi et al., 2025), leading to potential failures in high-stakes applications.

This unsupervised nature manifests in three critical failures that directly compromise safety in high-stakes domains. First, we confront a *hyperparameter sensitivity crisis* in which current UQ scores fluctuate drastically based on hyperparameter (Cecere et al., 2025). In practice, this sensitivity renders methods impractical for deployment since optimal parameters remain unknown due to rigid downstream constraints. This might mask inherent instability and create an illusion of safety based on parameter overfitting. Second, the field remains trapped in an *internal evaluation cycle* that fundamentally conflates self-consistency with correctness, which fails to detect "confident hallucinations" and

leads to a false decision in high-stakes domains. Third, we face a fundamental *lack of ground truth* that creates a recursive "judge problem" for UQ (Liu et al., 2025b). Since the "true uncertainty" of a model is inherently unobservable, evaluation methodologies must rely on the correlation with answer correctness as a proxy metric, yet obtaining objective correctness labels for open-ended tasks suffers from the exact same absence of ground truth. In high-stakes deployment, this circular dependency invalidates safety guarantees because we are effectively attempting to validate a critical system using a ruler that is just as elastic and unstable.

To resolve the problem, we argue that UQ researchers must abandon the pursuit of better unsupervised heuristics in favor of *supervised guarantees*. Specifically, the community should adopt a three-pillar roadmap to bridge the gap between internal belief and external reality. First, researchers should replace average-case benchmarks with *worst-case robustness* evaluations that explicitly stress-test the model on confident hallucinations (Carlini et al., 2022). Second, instead of merely optimizing evaluation performance, researchers should implement mechanism changes by training models with native uncertainty or deploying downstream applications with uncertainty (Quach et al., 2023; Gui et al., 2024; Chen et al., 2026a). Third, they should anchor uncertainty quantification in *objective truth* through atomic fact verification (Xie et al., 2025; Zheng et al., 2025) to eliminate the dependency on unstable model-based judges. By taking these collective actions, the field can move beyond unstable clustering and ensure that UQ serves as a true proxy for factual correctness, leading to preventing the deployment of overconfident models and ensuring that AI systems in critical domains operate with reliability.

## 2. Why Mainstream UQ is "Clustering": A Mechanistic View

### 2.1. Semantic Entropy: Explicit Clustering

Semantic Entropy (SE) (Kuhn et al., 2023) stands as a foundational technique in UQ for LLMs, because of its ability to disentangle linguistic variety from different sampling generations. Within the framework, SE and its variants: Semantic Alphabet Estimation (SAE) (McCabe et al., 2025), Semantic Energy (SEN) (Ma et al., 2025), Kernel Language Entropy (KLE) (Nikitin et al., 2024), Semantic Nearest Neighbor Entropy (SNNE) (Nguyen et al., 2025), and Semantically Diverse Language Generation (SDLG) (Aichberger et al., 2025) represent the most explicit implementation of the clustering paradigm in current research. There are other UQ methods for LLMs that integrate token-level probability and multiple sampling, which also fall into our claim (Vashurin et al., 2026; Cao et al., 2026).

**The Mechanism.** In standard generation, the sample space

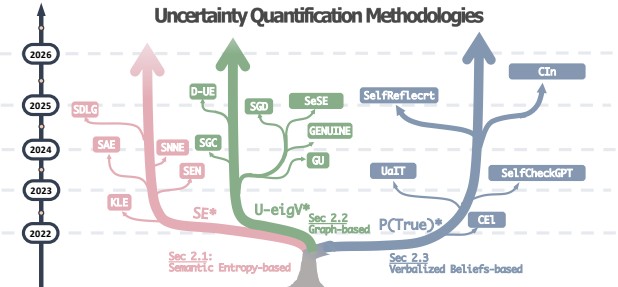

*Figure 1.* The common UQ methods for LLM and its representative work (name with *) for inductive discussions in Section 2.

is the vast set of possible token sequences. However, SE argues that this is the wrong level. Instead, it aggregates sequences that share the same meaning into classes, effectively treating each semantic cluster $C_i$ as a distinct *"Answer Class"*. Each answer class has a unique semantic meaning (e.g., "Paris" vs. "The capital of France" will be in the same Answer Class). The method generates sampled sequences $\mathcal{S}$, groups them into clusters $C_1, \ldots, C_M$ using an Natural Language Inference (NLI) model (He et al., 2021), and then calculates the entropy over these induced classes $U_{\text{SE}}(C|x) = -\sum_{i=1}^{M} p(C_i|x) \log p(C_i|x)$ Here, $p(C_i|x)$ represents the output probability of the $i$-th unique Answer Class. By treating semantic clusters as discrete categories, SE effectively transforms the uncertainty quantification problem from estimating the density of a continuous generation into a discrete classification problem over these derived answer classes. A lower entropy value implies that the model's probability mass is concentrated on a single Answer Class, while a higher value implies distribution across multiple conflicting Answer Classes.

**Why it is Clustering.** Mechanistically, Semantic Entropy acts as an explicit clustering algorithm that discretizes the model's output distribution. It maps the space of generated text to a discrete set of semantic clusters. The NLI model functions as the clustering criterion, determining class membership, while the entropy calculation measures the purity of these clusters. Consequently, the validity of the uncertainty score is bound by the quality of this clustering. A robust clustering correctly consolidates linguistic variations into a single semantic root, whereas a bad clustering fractures a single consistent belief into multiple spurious groups, artificially inflating the estimated uncertainty.

### 2.2. Graph-based Quantification: Implicit Clustering

Graph-based UQ methods (Lin et al., 2023) can be broadly interpreted as performing *implicit clustering* over a graph induced by sampled responses (Da et al., 2024a), such as approaches that Star Graphs Connectivity (SGC) constructs response-response or claim-response graphs and quantify uncertainty via connectivity patterns (Li et al., 2025), Graph Uncertainty (GU) uses graph centrality over claim-response

bipartite structures (Jiang et al., 2024), as well as semantic graph density (SGD) (Li et al.), hierarchical structural entropy (SeSE) (Zhao et al., 2025), multi-level graph aggregation (GENUINE) (Wang et al., 2025), or directional entailment relations (D-UE) (Da et al., 2024a). In these approaches, for example, uncertainty can be characterized by the degree to which the response set fragments into multiple semantically coherent components. A common and principled instantiation of this uncertainty quantification idea is through *spectral analysis of the graph Laplacian* (U-eigV), where the spectrum encodes the effective number of semantic modes present in the response distribution.

**The Mechanism.** Given an input $x$, the model generates a set of $m$ responses $\mathcal{S} = \{s_1, \ldots, s_m\}$. Under the black box setting, since token-level probabilities or hidden representations are inaccessible, the method first computes pairwise semantic similarity scores $a_{j_1, j_2}$ between responses[1]. These scores define a weighted similarity graph with adjacency matrix (Da et al., 2024a; Lin et al., 2024):

$$W = (w_{j_1, j_2}), \quad w_{j_1, j_2} = \frac{a_{j_1, j_2} + a_{j_2, j_1}}{2} \tag{1}$$

Let $D$ be the diagonal degree matrix with $D_{j_1, j_1} = \sum_{j_2} w_{j_1, j_2}$. The normalized graph Laplacian is then defined as:

$$L = I - D^{-1/2} W D^{-1/2}. \tag{2}$$

Uncertainty is quantified through the eigenvalues $\lambda_1 \leq \lambda_2 \leq \cdots \leq \lambda_m$ of $L$ via:

$$U_{\text{EigV}} = \sum_{k=1}^{m} \max(0, \, 1 - \lambda_k) \tag{3}$$

which can be interpreted as a *continuous estimate of the number of semantic meanings* present in the response set.

**Why this is clustering.** This procedure is a direct instantiation of *spectral clustering*, albeit without explicit cluster assignment (Ng et al., 2001). A classical result in spectral graph theory states that, for an unweighted graph, the multiplicity of the zero eigenvalue of the Laplacian equals the number of connected components (Von Luxburg, 2007). Thus, if the adjacency matrix $W$ encoded exact semantic equivalence, counting the number of near-zero eigenvalues would be equivalent to counting semantic clusters.

In practice, $W$ is dense and weighted, yielding a single connected component. However, spectral clustering relies on the distribution of small eigenvalues and eigen-gaps to infer an *effective number of clusters*. From this perspective, $U_{\text{EigV}}$ functions as an *internal cluster-validity index*: larger

---

[1]Such as using an external NLI model (He et al., 2020).

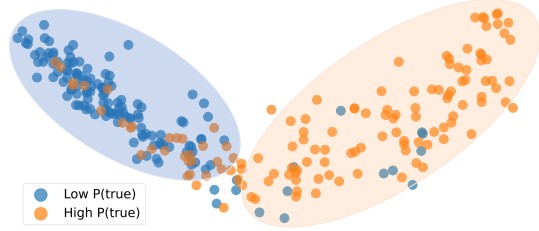

*Figure 2.* PCA visualization of Qwen2.5-32b-Instruct hidden states during P(true) estimation on the QASC dataset. The visualization demonstrates that the model's internal states during P(true) are geometrically partitioned into distinct belief clusters, empirically validating that P(true) functions as an implicit clustering.

values indicate that the response mass fragments into multiple coherent semantic modes, while smaller values indicate concentration around a single dominant mode. Conversely, when responses express multiple incompatible or weakly related meanings, many eigenvalues remain small, producing a substantially larger uncertainty score.

### 2.3. Verbalized Beliefs as Latent Confidence Clustering

P(true) (Kadavath et al., 2022), which is the most widely used verbalized uncertainty quantification method, asks LLMs to explicitly assess the correctness of their own generated answers and interprets the resulting confidence as an uncertainty signal. Subsequent work extends this paradigm by exploring different forms of confidence elicitation and self-reflection, including Confidence Introspection (CIn) (Xi et al., 2026), Confidence Elicitation (CEl) (Xiong et al., 2023), SelfCheckGPT (Manakul et al., 2023), UaIT (Liu et al., 2024), and SelfReflect (Kirchhof et al., 2025), which vary in prompting strategies or training procedures but all rely on the model's internally expressed confidence regarding a generated response. In this paradigm, the model is repurposed to evaluate the density of its own generation.

**The Mechanism.** For the purpose of induction, we take P(true) as a standing example. The core operation of P(true) is to probe the model's local confidence. The method appends a verification prompt to a generated answer $\hat{y}$ (e.g., "Is the proposed answer true?") and extracts the probability assigned to the token "True". Formally, the confidence score is defined as $U_{\text{P(true)}}(x, \hat{y}) = 1 - P(\text{"True"}|x, \hat{y})$. This value represents the model's scalar estimation of validity based on its immediate next-token distribution. Effectively, P(true) acts as a point-wise probe of the model's conviction at the specific coordinate of the generated answer.

**Scope Clarification.** Before proceeding, we note that the clustering mechanism we identify for P(true) differs mechanistically from the explicit semantic clustering of SE (Section 2.1) or the spectral clustering of graph-based methods (Section 2.2). Rather than clustering *between multiple generations*, P(true) partitions the model's *own latent space* into confidence regions and tests whether a generated answer

falls within a high-belief region.

**Why it is Clustering.** This method utilizes the LLM as a soft clustering function that defines regions of high-confidence concepts within its latent space. When we query P(true), we are performing a membership test against these *internal confidence clusters*. A high probability score indicates that the generated answer lies geometrically close to the centroid of the model's internal representation. Conversely, a low score marks the sample as an outlier far from the cluster center. We show the evidence in Figure 2. The visualization shows that high P(true) (low uncertainty) samples form a dense cluster that is geometrically separated from the low P(true) samples, empirically confirming that P(true) is an implicit clustering of internal beliefs. Therefore, the LLM is not judging factual correctness; it is actually calculating the geometric distance between the generated output and the center of its own parametric confidence.

### 2.4. Methods Outside Our Framework

We do not claim that every conceivable UQ method falls within this framework. Below we delineate representative methods that operate under different paradigms.

**Token-level entropy and perplexity** The most direct approach is to compute entropy or perplexity over the token distribution at generation time. While free from the clustering machinery we critique, these methods have been shown to perform poorly on LLM free-form generation tasks (Kuhn et al., 2023), which motivated the development of Semantic Entropy and its successors. The field's migration away from raw token-level methods toward multi-sample semantic aggregation is itself evidence that the clustering paradigm has become the mainstream.

**Ensemble-based methods** Classical UQ approaches such as Deep Ensembles (Malinin & Gales, 2020) capture uncertainty by aggregating multiple independently trained model copies. These methods sit outside our framework because they measure disagreement across models rather than internal consistency within a single model. However, training and serving multiple independent copies of a modern LLM is prohibitive for most practitioners, which is why the community has largely abandoned this direction in favor of single-model heuristics.

**Supervised approaches** A small but growing line of work trains supervised classifiers on labeled data to predict correctness, such as Azaria & Mitchell (2023), which uses internal hidden states with ground-truth correctness labels. This approach is precisely the externally grounded, supervised paradigm we advocate for in Section 5 later. Rather than measuring internal consistency, these methods anchor uncertainty to actual correctness labels, breaking the unsupervised loop we critique throughout this paper.

| Method Pair | 10% | 20% | 30% |
|---|---|---|---|
| $U_{\text{SE}}$ vs $U_{\text{EigV}}$ | 0.134 | 0.166 | 0.266 |
| $U_{\text{SE}}$ vs $U_{\text{P(true)}}$ | 0.080 | 0.159 | 0.229 |
| $U_{\text{EigV}}$ vs $U_{\text{P(true)}}$ | 0.224 | 0.319 | 0.404 |

*Table 1.* Jaccard similarity among Top-k% highest-uncertainty samples identified by different UQ methods on QASC using Qwen2.5-32B. Low overlap indicates fundamental disagreement on which instances are "uncertain."

## 3. The Diagnosis: Why Clustering Fails

In clustering, internal validity indices measure how well-separated clusters are, but cannot guarantee that the clusters map to real-world semantics (Vinh & Houle, 2010). We observe identical pathologies in modern UQ research.

### 3.1. The Parameter Sensitivity Crisis

Although mainstream UQ methods achieve relatively high scores on established internal metrics, these methods function as tunable heuristics rather than principled measurements of epistemic states. Similar to clustering algorithms, UQ results exhibit fragile sensitivity to human-specified parameters and assumptions, which applies to multiple sampling UQ methods. In clustering, this dependency is well-documented: the choice of K in K-means (Kodinariya et al., 2013), the distance metric (Aggarwal et al., 2001), or the algorithm itself (Rodriguez et al., 2019) can yield entirely different cluster assignments from identical data. No internal validity index can adjudicate which configuration captures "true" structure, because no ground truth exists.

UQ methods suffer from analogous pathologies. First, different uncertainty quantification approaches produce incomparable outputs. Each operates on a different scale, making it difficult to interpret or compare raw uncertainty scores across methods (Vashurin et al., 2025). Even when we normalize for comparison by selecting fixed percentiles of the highest-uncertainty samples, the disagreement persists. As shown in Table 1, we compute the Jaccard similarity among the top 10%, 20%, and 30% highest-uncertainty samples identified by each method. The overlap remains low, suggesting that different approaches disagree fundamentally on which instances should be considered "uncertain."

Second, even within a single method, parameter choices substantially alter results. Prior work has shown that varying the number of sampling generations $n$ directly affects the stability and magnitude of uncertainty estimates (Kuhn et al., 2023). Similarly, the NLI threshold used to determine semantic equivalence (Farquhar et al., 2024), temperature scaling parameters (Cecere et al., 2025), and prompt formulation (Gao et al., 2024) all modulate uncertainty estimates in ways that lack external justification.

This parameter dependence reveals a deeper issue: UQ's internal metrics cannot validate whether uncertainty esti-

mates reflect genuine uncertainty states. High performance can arise from incidental correlations between parameter settings and task performance, without implying semantic validity of the uncertainty estimates.

### 3.2. The "Internal Evaluation" Trap

This pathology applies broadly to any UQ method that operates without grounding in external truth, whether single-sample or multi-sample. Current evaluation norms predominantly rely on metrics that reward *self-consistency*, operating on the tacit assumption that truth corresponds to the mode of the model's generation distribution. We argue that this assumption is fundamentally flawed due to the phenomenon of *"confident hallucination."* Large language models often exhibit high determinism in their errors (Simhi et al., 2025). In such cases, a model is consistently wrong, rendering clustering-based metrics deceptive. High consistency merely indicates that the model has converged to a stable state, not necessarily a factual one.

We contend that the various uncertainty quantification metrics are philosophically identical despite their mathematical differences. They all function as clustering mechanisms that assess the internal consistency of generated outputs rather than their alignment with external reality. This process is analogous to calculating a Silhouette coefficient, where a high score simply indicates that data points are tightly grouped together, regardless of whether the cluster itself is meaningful or correct. Therefore, these methods share a single fatal flaw because they rely on the assumption that internal stability is a proxy for truth.

### 3.3. The Lack of Ground Truth

This limitation applies universally to all UQ methods, regardless of whether they rely on multi-sample or single-sample generation, or verbalized confidence. Perhaps the most fundamental limitation is what we term the "judge problem": for any given sample, who defines how uncertain the model should be? When a model claims "80% confidence," how do we verify whether this figure reflects a genuine uncertainty state rather than a mathematical artifact? Unlike classification accuracy, where ground truth labels provide an objective standard, uncertainty has no direct ground truth. We cannot observe the *"true uncertainty"* a model should express (Beigi et al., 2024).

Current evaluation pipelines attempt to circumvent this problem by establishing a proxy relationship: high uncertainty should indicate low correctness, and vice versa. Metrics such as Area Under the Receiver Operating Characteristic curve (AUROC) (Lin et al., 2023) operationalize this assumption, treating the uncertainty-correctness correlation as a stand-in for semantic validity.

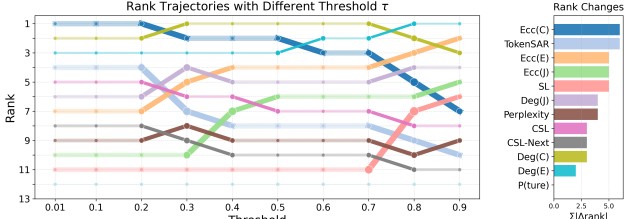

*Figure 3.* The effect of correctness threshold $\tau$ on UQ method evaluation consistency. As the threshold varies, method rankings become unstable. Figure adapted from Liu et al. (2025b).

However, this proxy itself rests on unstable foundations, particularly for open-ended generation tasks. Obtaining accurate correctness labels is inherently challenging because correct answers are not unique. Semantically equivalent responses may differ substantially in surface form. Different evaluation approaches yield different correctness judgments: semantic entropy relies on RougeL scores exceeding 0.3 between generations and reference answers (Kuhn et al., 2023), while more recent work employs another LLM to serve as the "correctness function" (Lin et al., 2024; Da et al., 2025; Chen et al., 2025). Yet these judges constitute another layer of heuristic approximation. As Liu et al. (2025b) demonstrates, such correctness functions are noisy, expensive, and biased. As shown in Figure 3, flaws in the correctness function propagate through the entire evaluation pipeline, distorting metrics like AUROC and causing significant instability in UQ method rankings. When the judge itself is unreliable, any verdict on the uncertainty quality becomes suspect.

This situation is analogous to calibrating a spring scale using a rubber band; without a fixed reference point, measurement loses meaning. The UQ community currently lacks external validation protocols comparable to the Adjusted Rand Index or Normalized Mutual Information in clustering evaluation. We have no mechanism to verify whether our "uncertainty estimates" align with any external reality, leaving us trapped in a cycle of internal validation that, however mathematically sophisticated, cannot guarantee semantic validity.

## 4. Alternative Views

**Objection 1: Sensitivity is a Feature for Uncertainty.** *Argument:* Practitioners might argue that the sensitivity of UQ metrics to hyperparameters (e.g., sampling temperature, NLI thresholds) is a necessary degree of freedom, not a pathology. These parameters allow engineers to calibrate the system for specific risk tolerances (trading off precision for recall). In this view, variation in UQ scores is not evidence of fragility, but evidence that the method is responsive to generation dynamics.

*Response:* We disagree with the characterization of hyperparameter sensitivity as a desirable degree of freedom. This perspective inaccurately conflates the necessary selection of

a post-hoc decision threshold with the pathological volatility of the underlying scoring mechanism. While varying a decision threshold allows for legitimate trade-offs between precision and recall, a UQ metric that fluctuates drastically based on generation parameters indicates a lack of robustness rather than adaptability. Such sensitivity compromises the integrity of comparative evaluation because it incentivizes the reporting of hyper-optimized configurations that mask the method's inherent instability. This creates an illusion of technical progress where there is only parameter overfitting. Furthermore, relying on precise hyperparameter tuning renders these methods impractical for real-world deployment since the optimal generation parameters are often dictated by downstream task constraints or remain unknown due to distribution shifts (Podkopaev; Flovik, 2024). A rigorous UQ metric must demonstrate invariance across the reasonable operating range of the underlying model to ensure that the measured confidence reflects the model's internal knowledge state rather than artifacts of the decoding strategy (Song et al., 2025; Cecere et al., 2025).

**Objection 2: Uncertainty Measures Belief, Not Truth.** *Argument:* From a strict Bayesian perspective, uncertainty quantification aims to faithfully represent the posterior distribution of the model's parameters given the data. Critics may argue that if a model has converged to a coherent (albeit incorrect) belief state due to biased training data, a "good" UQ metric should accurately report high confidence. In this view, the failure lies in the model alignment, not the UQ metric. By demanding that UQ metrics detect hallucinations, the authors are conflating calibration to belief with calibration to reality.

*Response:* We argue that for real-world safety, the utility of a UQ metric depends entirely on its ability to distinguish correct outputs from incorrect ones (Manakul et al., 2023; Yang et al., 2023). While separating a model's internal belief from objective truth is theoretically valid, strictly following this distinction in practice produces metrics that might be useless for risk mitigation (Devic et al., 2025). If a metric faithfully reports that a model is confident in a hallucination, it effectively acts as an accomplice to the error rather than a safeguard. We certainly acknowledge that measuring the model's subjective belief serves a valuable purpose in research, allowing scientists to understand the model's internal state. However, we contend that practical utility must take precedence for trustworthy deployment. (Patel et al., 2026) has shown that while internal features in LLMs reliably track prediction correctness, suppressing these features does not affect the output.

Besides, the community's dominant evaluation metrics already implicitly treat UQ as a measure of truth. AUROC and AUPRC are computed against answer correctness labels, meaning a method scores well only when high uncertainty corresponds to incorrect answers. The adoption of these metrics reflects a common-sense expectation that uncertainty should track factual correctness, not merely internal belief.

**Objection 3: Consistency is a Sufficient Proxy for Correctness.** *Argument:* Critics may argue that our distinction between consistency and correctness is theoretically valid but practically negligible. Extensive empirical evaluations in prior works (Kuhn et al., 2023; Da et al., 2024a; Lin et al., 2023) demonstrate that internal consistency metrics achieve high AUROC scores on standard benchmarks, effectively distinguishing between correct and incorrect answers. Therefore, existing internal metrics and clustering-based methods already serve as adequate proxies for reliability.

*Response:* We refute the assertion that internal consistency is a sufficient proxy for correctness. High aggregate scores such as AUROC are often misleading because they are dominated by easy examples where the model is appropriately uncertain about its errors (Santilli et al., 2025). However, the correlation between consistency and correctness breaks down precisely where reliability is most critical, particularly during confident hallucinations. In these cases, the model consistently outputs the same incorrect answer due to mimetic errors or mode collapse, meaning consistency metrics merely measure the stability of the error rather than its validity (Kalavasis et al., 2025). A heuristic that filters out a majority of obvious failures is inadequate, especially in high-stakes applications (Machcha et al., 2025). Therefore, relying solely on consistency creates a survivor bias where the most dangerous errors, specifically those that appear stable and confident, pass through undetected.

**Objection 4: Ground Truth is Intractable for Generative Tasks.** *Argument:* Perhaps the most practical objection is that for open-ended generation (e.g., creative writing), a single "Ground Truth" simply does not exist. Truth in language is often plural and subjective. Critics argue that demanding external validation effectively restricts UQ research to toy problems (like Multiple Choice problem) and ignores the generative nature of LLMs. Therefore, internal consistency is the only scalable signal available.

*Response:* While we acknowledge that open-ended generation admits valid linguistic variation, this objection fundamentally conflates surface-level diversity with informational correctness. In high-stakes domains such as medical diagnosis or legal precedent analysis, the underlying atomic claims possess objective truth values regardless of the syntactic structure used to express them. A legal argument may be constructed through various rhetorical strategies, yet the cited case law is either hallucinated or factual; similarly, a medical treatment plan may vary in tone, but the prescribed dosage is either safe or toxic (Kim et al., 2025). We contend that hiding behind the philosophical complexity of language

to justify the absence of external validation is methodologically unsound for safety-critical systems. True reliability demands that we disentangle the subjective manner of answers from the objective validity of the content, ensuring that UQ measures the system's adherence to reality, as explored in (Liu et al., 2026). In other words, it is hard to track the ground truth in open-ended generation, but it is the task our researchers need to pursue.

**Objection 5: Scaling Solves Reliability.** *Argument:* Some proponents argue that the calibration problem is transient and will resolve itself with scale. Empirical studies (Kadavath et al., 2022) suggest that larger models are naturally better calibrated than smaller ones. Therefore, instead of developing complex UQ methods, the community should simply focus on scaling up model parameters and training data to achieve reliable confidence estimates.

*Response:* While scaling laws improve general task performance, empirical evidence indicates that the alignment techniques required for deployment, specifically Reinforcement Learning from Human Feedback (RLHF), actively push model distributions away from true probabilities to match human preferences (Kadavath et al., 2022; Achiam et al., 2023; Tian et al.). Optimizing for human preference encourages models to adopt an authoritative tone regardless of factual accuracy, effectively suppressing necessary expressions of doubt. Consequently, larger models do not automatically become more reliable, and they often become merely more persuasive in their errors. This phenomenon exacerbates the clustering pathology we identify in this work, as a scaled-up model is capable of generating highly coherent and internally consistent hallucinations that are statistically indistinguishable from truth when viewed solely through internal metrics. Relying on parameter scaling is therefore insufficient, as it amplifies the model's ability to maintain a consistent narrative without addressing the fundamental misalignment between internal geometric consensus and external reality.

# 5. The Path Forward: From Unsupervised Heuristics to Supervised Guarantees

To transition UQ from clustering to a rigorous science of external verification, we propose a three-pillar paradigm shift: evaluation, mechanism and grounding.

## 5.1. Evaluation: From Average Performance to Worst-Case Robustness

Current evaluation norms in UQ largely depend on aggregate metrics calculated over standard benchmarks. While useful for general performance monitoring, these metrics fail to capture the catastrophic failure modes critical for safety. We propose shifting the evaluation paradigm from maximizing average-case separation to ensuring worst-case robustness, considering sensitivity to hyperparameters.

### 5.1.1. TAIL-RISK EVALUATION

Standard metrics such as AUROC are often dominated by the vast majority of samples where the model behaves predictably. The metric is inflated because existing methods can easily distinguish between correct answers with high confidence and incorrect answers with low confidence. This statistical separation allows methods to achieve high AUROC scores on the dataset level.

However, this statistical aggregate might hide critical failures at the **instance level**. In practical deployment, users do not rely on the model's average performance over a dataset; they make high-stakes decisions based on specific, individual queries. A UQ method that achieves 0.80 AUROC globally but assigns high confidence to a specific factual error has failed in its primary safety objective. The global metric masks this local failure because the error is statistically drowned out by the volume of easy samples.

**First Principle of Safety: Vulnerability over Average** To rigorously define this evaluation gap, we invoke the *First Principles* of Membership Inference Attacks (MIA) established by Carlini et al. (2022). In privacy auditing, they argue that privacy is not an average-case metric. An attack that successfully identifies 0.1% of training set members with high confidence is a catastrophic privacy breach, even if its average accuracy across the entire population is equivalent to random guessing. The failure of the system is defined by its outcome on the most vulnerable samples (High-Leverage Points), not the average sample.

We posit that UQ is isomorphic to MIA. Just as a system is not private if it leaks a single user's data, a system is not safe if it validates a single high-risk hallucination. Similar to MIA, the high-confidence responses (which users tend to trust implicitly) and the low-confidence responses (which users are expected to discard) are important.

One immediate improvement, considering the critical data, to current evaluation norms would be to calculate AUROC exclusively on these high-uncertainty and/or low-uncertainty subsets, rather than the full dataset. While this subset AUROC would reduce the inflation caused by the volume of easy samples, it remains a ranking metric that does not provide a hard safety guarantee.

To rigorously audit reliability, we must go a step further and explicitly define the UQ module as an active warning system (or Rejection Mechanism) (Barandas et al., 2022). In this operational view, the system does not merely output a score, but makes a binary decision: to *Accept* (remain silent) or *Reject* (trigger an alert). Only by framing UQ

as a binary classifier of errors can we adopt the *True Positive Rate (TPR) at Low False Positive Rate (FPR)* metric established in privacy auditing (Carlini et al., 2022). Here, a "False Positive" represents a false alarm on a correct answer. Therefore, we must fix the FPR to a strictly low threshold (e.g., $< 0.1\%$) and measure the TPR. This metric shows whether the warning system still successfully catches the catastrophic confident hallucinations at the limit of FPR.

### 5.1.2. SENSITIVITY REPORTING

As diagnosed in Section 3.1, many current quantification metrics function as fragile heuristics that fluctuate aggressively with hyperparameters. We argue that the prevailing practice of reporting peak performance after exhaustive hyperparameter tuning is methodologically equivalent to p-hacking, as it obscures the operational fragility of the underlying mechanism. To rigorously assess whether a method measures genuine reliability rather than decoding artifacts, we advocate for the *Area Under the Stability Curve (AUSC)* as a mandatory reporting standard.

Mechanistically, the AUSC represents the integral of a performance metric, typically AUROC, across a continuous sweep of a hyperparameter. Rather than presenting a single scalar derived from an optimal configuration, researchers need to demonstrate the performance profile over the entire feasible parameter space. This approach is necessary to expose the structural limitations of consistency-based methods. For instance, consider the behavior of a UQ across the sampling temperature $T \in [0, 1.0]$. In the regime where $T$ approaches zero, the generation becomes deterministic, artificially suppressing the variance required by sampling-based algorithms; consequently, the AUROC frequently collapses to random guessing (0.5) because the method cannot detect divergence in a deterministic output. As the temperature increases, the injection of stochasticity allows the method to distinguish between stable and unstable generations, increasing the AUROC. However, a method that achieves state-of-the-art separation only within a narrow thermal window (e.g., $T = 0.7$) while failing at adjacent values implies that the signal is an artifact of specific decoding dynamics rather than a true property of the model's knowledge. The idea of AUSC shows that a valid quantification method must be *distributionally robust*. The assessment of risk should remain stable despite variations in the parameters.

### 5.2. Mechanism: From Post-hoc Heuristics to Native Guarantees

Having established that current evaluation protocols often mask the fragility of heuristic scores, we now turn to the generative mechanisms themselves. We must move beyond the current paradigm of passive interpretation. We argue that the focus must shift from interpreting output distributions to engineering uncertainty directly into the system architecture.

### 5.2.1. CONFORMAL PREDICTION AS THE APPLICATION

Raw confidence scores lack physical grounding, as we demonstrated in Section 2, they quantify the geometric compactness of a semantic cluster rather than the probability of correctness. A score of 0.85 is meaningless if it merely reflects that the model is stubbornly consistent in its error. Consequently, we argue for evaluating quantification methods not in isolation, but through their utility in rigorous downstream applications, specifically *Conformal Prediction (CP)* (Quach et al., 2023; Su et al., 2024). By transforming vague uncertainty scores into prediction sets that contain the true answer with a user-specified probability, CP forces the uncertainty estimate to confront reality. We view CP as an evaluation framework that exposes information through the resulting prediction sets.

In this context, the *Efficiency (Set Size)* of the conformal set serves as a critical, truth-aware metric. Consider two UQ methods A and B used as nonconformity scores at 90% target coverage. Both are guaranteed the same coverage, but if A assigns high confidence to hallucinated answers, it must include more candidates to maintain coverage, resulting in larger sets. By comparing set sizes at equal coverage, we obtain a truth-aware comparison that penalizes methods for confident hallucinations. While a heuristic score might assign high confidence to a hallucination, a valid conformal predictor operating under a strict coverage constraint is mathematically forced to expand the prediction set to include the ground truth. In the extreme case, "confident hallucination" manifests as a *Set Size Explosion*, where the prediction set grows to encompass a massive portion of the vocabulary. A system that requires the entire dictionary to guarantee coverage is functionally useless. Set size at equal coverage thus offers a diagnostic property that internal consistency metrics fundamentally lack.

### 5.2.2. POST-TRAINING FOR NATIVE UNCERTAINTY

The fundamental limitation of the strategies discussed thus far is their retroactive nature, and they attempt to extract signal from a model that was never trained to express uncertainty (Heo et al.). Consequently, we argue that the community must pivot from standard Instruction Tuning to a rigorous practice of Uncertainty Alignment.

This paradigm shift requires utilizing Post-Training stages, such as Reinforcement Learning from Human Feedback (RLHF), to explicitly incentivize the articulation of confidence levels (Lin et al., 2022). The training objective should reward the accurate prefacing of generations with granular verbal markers (e.g., "I am confident that..." versus "It is possible that...") (Stangel et al., 2025; Ulmer et al., 2025; Zhang et al., 2026). Mechanistically, this process reorga-

nizes the latent space by optimizing for these distinct linguistic headers on Out-of-Distribution data. We anchor the model's internal representations directly to explicit validity claims. This transforms uncertainty from a latent geometric artifact, which current methods struggle to interpret without supervision, into a transparent, communicated feature of the generation itself.

### 5.3. Grounding: Establishing Objective Truth

We conclude our structural critique by addressing the absence of objective verification in current methodologies. To resolve the unsupervised nature of existing UQ, the field must break the closed loop of using models to judge themselves and instead anchor evaluation in external reality.

#### 5.3.1. MANDATORY "UNIT TESTING"

Evaluating quantification methods on open-ended creative generation is theoretically unsound because the calculation of rigorous metrics like AUROC and TPR at low FPR requires a binary definition of failure. These critical safety metrics are mathematically indeterminate in subjective domains where the ground truth is fluid or debatable. We argue that any quantification method must pass **Gold-Standard Unit Tests** in verifiable environments before being applied to open-ended generation.

The necessity of this standard arises from the requirement for absolute labels to validate the separation capability of metrics like AUROC. We define a verifiable environment as any setting where the validity of an output can be algorithmically determined without reliance on another language model (Yao et al., 2022). This environment contains, but is not limited to, code generation benchmarks (Chen, 2021; Jimenez et al.) where correctness is proven by execution, and mathematical reasoning datasets (Hendrycks et al.) where the final answer is a fixed constant. If a method fails to correlate confidence with correctness in these deterministic settings, where the distinction between truth and falsehood is unambiguous, it lacks the credibility to judge the reliability of open-ended tasks.

#### 5.3.2. ATOMIC FACT VERIFICATION

While verifiable environments serve as the initial filter, we must extend this rigor to the open-ended domains. Relying on another large language model to score reliability in these unstructured contexts constitutes circular reasoning, as it essentially validates one model's hallucinations with another model's biases. We argue for replacing subjective scoring with Atomic Fact Verification (Xie et al., 2025; Zheng et al., 2025), which acts as a rigorous decomposition standard. This protocol mandates the decomposition of complex narratives into atomic claims that function as indivisible units of information. Verification must then proceed through di-

verse external authorities tailored to the claim type. This includes not only cross-referencing against search engines and structured knowledge bases but also the integration of formal theorem provers like Lean4 (Moura & Ullrich, 2021) for logical validity and the utilization of deep search agents capable of performing multi-hop evidence retrieval. This approach transforms the label space from consistency to objective factuality, providing the external validation that is required to truly benchmark the performance of quantification methods against reality.

## 6. Conclusion

This position paper concludes that current UQ for LLMs fundamentally operates as unsupervised clustering, measuring internal consistency rather than external factual correctness. We have demonstrated that this structural limitation renders existing methods blind to confident hallucinations, thereby creating a deceptive sense of safety in high-stakes deployments. By identifying the critical pathologies, we argue that the community must shift from unsupervised heuristics toward a supervised guarantee framework. This proposed paradigm shift is essential to transform the uncertainty of models into a reliable proxy for reality and ensure the trustworthy integration of LLMs into society.

## Acknowledgment

The work was partially supported by NSF award #2442477 and #2550203. We thank Amazon Research Awards, Cisco Faculty Research Awards, and Toyota Faculty Research Awards. The authors acknowledge Google and OpenAI for providing us with API credits and Research Computing at Arizona State University for providing computing resources. The views and conclusions in this paper are those of the authors and should not be interpreted as representing any funding agencies.

## Impact Statement

This paper presents a critique of current research practices in Uncertainty Quantification for Large Language Models. If adopted, our recommendations on evaluation, grounding, and mechanism could lead to the more robust deployment of AI systems in critical fields like healthcare and law. Conversely, continuing to rely on internal consistency heuristics poses a severe risk of deploying overconfident models that fail silently in high-stakes scenarios, creating a deceptive sense of safety for end-users.

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
