# OpenReview forum: "Position: Uncertainty Quantification in LLMs is Just Unsupervised Clustering"
_ICML.cc/2026/Position_Paper_Track — ICML 2026 Position Paper Track regular_

### Official Review · Reviewer_YeSq · 2026-03-03

**Significance:** 3
**Argument Clarity:** 3
**Rating:** 4
**Confidence:** 2

**Questions:**

Under weaknesses I posted some comments that can be interpreted as questions.

In addition, I have a question about AUROC, which is multiple times discussed in the paper. I did not understand how AUROC can be used as a metric for LMMs, because it is a binary classification metric. What are the two classes here?

**Alternative Views Section:**

Yes

**Compliance With Llm Reviewing Policy A Conservative:**

Affirmed.

**Discussion Potential:**

3

**Paper Summary:**

This paper argues that existing uncertainty quantification methods for LMMs in essence perform clustering of the output space. Three groups of methods are considered: methods based on semantic entropy, methods based on graphs, and verbalized uncertainty quantification methods. For all three groups of methods, the authors provide informal arguments why these methods reduce to unsupervised clustering. After discussing alternative views, the authors suggest some possibilities to move forward, such as analyzing worst-case robustness, providing statistical guarantees or collecting uncertainty-labeled datasets.

**Position:**

Yes

**Position In Title:**

Yes

**Related Work:**

3

**Strengths And Weaknesses:**

Strengths:
- This is a well-written paper that discusses a timely and important problem.
- As a survey, I think that this paper is a very good introduction to the topic of uncertainty quantification for LLMs. I did learn some new things by reading this paper.

Weaknesses:
- Section 5 is for me the weaker part of the paper. For conformal prediction, I don't see how it can provide a solution for the unsupervised clustering problem of the methods discussed in Section 2. For conformal prediction one needs labeled calibration data, but it has been argued in the beginning of the paper that labelled data is usually not available for LMMs. So, this is a paradox. Furthermore, for applying conformal prediction to LMMs one will eventually need some type of clustering, too, because a query does not have a single correct answer. Probably, one needs structured versions of conformal prediction, such as conformal risk control. These aspects are not discussed in-depth in the paper.
- What I also find a bit weird in Section 5 is the framing of establishing objective truth as future work. Isn't this already happening in the LMM community? I am not working on LMM topics, but I did see at recent conferences several papers that evaluate uncertainty in LMMs using labeled datasets (e.g. where end users give scores to the answers of LMMs). I don't remember the authors of those papers, but I would think that this is already becoming the SOTA way of evaluating the uncertainty of LMMs. I did a quick search for papers that develop conformal prediction methods for LMMs. ChatGPT returns me more than ten papers, some of them as early as 2023. Conformal prediction needs labeled calibration datasets, so I would argue that the transition from unsupervised clustering to supervised evaluation of uncertainty in LMMs is already a mainstream idea.
- The authors don't make any formal distinction between aleatoric and epistemic uncertainty. However, I do think such a distinction is important for this paper. For example, I would argue that all clustering-based methods discussed in Section 2 focus on aleatoric uncertainty, but they do not consider epistemic uncertainty at all. Epistemic uncertainty is in essence what is discussed in Section 3.1, because parameter sensitivity and incorrect configuration of parameters are linked to this type of uncertainty. A quick search has learned that there are already papers that discuss aleatoric and epistemic uncertainty in the context of LMMs, see e.g. Hamidish et al. Uncovering Confident Failures: The Complementary Roles of Aleatoric and Epistemic Uncertainty in LLMs, Neurips 2025.

**Support:**

2

---

> ### Author Rebuttal · Authors · 2026-03-29
>
> We sincerely thank the reviewer for the thoughtful feedback and for finding our paper a 'very good introduction to the topic.' Here are our responses:
>
> > **W1: CP needs labeled calibration data, but the paper argues labeled data is unavailable. This is a paradox.**
>
> **Answer:** We believe this conflates two distinct types of "ground truth" in our paper. In Section 3.3, we argue that the **ground truth of uncertainty itself** ("how uncertain should the model be?") is inherently unobservable. This is the "judge problem."
>
> CP, on the other hand, requires labeled data about **answer correctness**, not about uncertainty states. In verifiable environments such as code generation (correctness via execution), mathematical reasoning (fixed numerical answers), and factual QA with knowledge base verification, such correctness labels are obtainable. These are exactly the "Gold-Standard Unit Tests" we advocate in Section 5.3.1.
>
> So there is no paradox. The unavailable ground truth is about uncertainty states. The available ground truth (in verifiable settings) is about answer correctness, which is what CP needs.
>
> The reviewer is correct that for open-ended generation, structured CP variants (such as conformal risk control) would be needed. We will expand Section 5.2 to discuss this.
>
> > **W2: Establishing objective truth is framed as future work, but isn't this already happening?**
>
> **Answer:** The dominant evaluation protocol already uses labeled data. Most UQ papers report AUROC against correctness labels.
>
> However, our argument in Section 5 is about **method design**, not evaluation alone. While evaluation uses correctness labels to *judge* UQ methods post-hoc, the UQ methods themselves are still designed as unsupervised heuristics operating without reference to ground truth at inference time. No current mainstream UQ method incorporates external factual verification into its scoring mechanism. Our roadmap advocates for change at the method level, through native uncertainty via post-training (Section 5.2.2), atomic fact verification in the UQ pipeline (Section 5.3.2), and CP as a framework forcing scores to confront reality (Section 5.2.1). We will revise Section 5 to make this evaluation-time vs. method-design-time distinction more explicit.
>
>
> > **W3: The paper does not distinguish between aleatoric and epistemic uncertainty.**
>
> **Answer:** We would like to offer context on why we did not adopt this decomposition.
>
> The aleatoric/epistemic distinction in LLM UQ remains unsettled. Liu et al. [1] note that the classical categorization does not straightforwardly apply to LLM generation. A meta-analysis by [2] finds that approximately 62% of existing UQ works for LLMs assess **total uncertainty** without distinguishing between types.
>
> Our paper deliberately sidesteps this unsettled decomposition in favor of a more operationally relevant distinction, **internal consistency vs. external correctness**. This cuts across the aleatoric/epistemic divide and directly addresses the practical failure mode we identify. Whether one labels the uncertainty as aleatoric or epistemic, the core problem remains the same. Current methods rely on internal signals and cannot detect systematic errors.
>
> We will add a discussion of how our framework relates to this decomposition, referencing Hamidish et al.  as suggested.
>
> [1] Liu, Xiaoou, et al. "Uncertainty quantification and confidence calibration in large language models: A survey." Proceedings of the 31st ACM SIGKDD Conference on Knowledge Discovery and Data Mining V. 2. 2025.
> [2] Lambert, Benjamin, et al. "Trustworthy clinical AI solutions: a unified review of uncertainty quantification in deep learning models for medical image analysis." Artificial Intelligence in Medicine 150 (2024): 102830.
>
>
>
> > **Q1: How is AUROC used for LLMs? What are the two classes?**
>
> **Answer:** In the UQ evaluation, AUROC treats the problem as a binary classification. The positive class is the set of samples where the model's answer is incorrect (determined by a correctness function such as RougeL above a threshold, or an LLM-as-judge). The negative class is the set of samples where the model's answer is correct. The uncertainty score serves as the prediction score. A good UQ method assigns higher uncertainty to incorrect answers, yielding a high AUROC. This is standard practice. Notably, the widespread use of AUROC implies that the community already implicitly adopts an objective view of uncertainty, where UQ scores should correlate with factual correctness. This is consistent with the position we advocate in our paper, and it is why we argue for a transfer of model design as we mentioned earlier.

---

> > ### Author Rebuttal · Reviewer_YeSq · 2026-04-01
> >
> > I thank the authors for their detailed answer.
> >
> > Concerning aleatoric and epistemic uncertainty: I am perfectly fine with your viewpoint, but please add a discussion in the paper if it would be accepted.
> >
> > Concerning ground-truth labels for CP in LMMs, perhaps my remark was a bit unclear. Attaching ground truth to the answer of a query as a whole is probably impossible in LMMs, but one can elicit ground truth for certain aspects of an answer by asking users specific questions, such as:
> > - Did the answer of the query correctly report historical facts?
> > - Are all the calculations made in the answer correct?
> > - Does the answer contain a political view?
> > That would be a way of collecting ground truth. As far as I know, it is already done in LMM research for the moment.
> >
> > I was positive about this paper. I will stick to my initial score.

---

### Official Review · Reviewer_oBYr · 2026-03-10

**Significance:** 2
**Argument Clarity:** 2
**Rating:** 2
**Confidence:** 4

**Questions:**

1. What do authors imply by predictive uncertainty? Do authors mean to define it objectively (a measure of UQ = 0 iff no mistake allowed)?

2. Does the position in the paper hold only for consistency-based measures, or for confidence-based (the ones that operate with probabilities of tokens/sequences) as well?

**Alternative Views Section:**

Yes

**Compliance With Llm Reviewing Policy A Conservative:**

Affirmed.

**Discussion Potential:**

3

**Final Justification:**

I thank the authors for their answers.

I saw the final sentence regarding continuing the discussion, but unfortunately, we cannot technically proceed with it. The discussion is limited only to two rounds of exchanges, and public comments are not visible to the authors.

Regarding W1, I kindly disagree with the authors' insistence on the generality of their position. Both variants proposed by the authors, such as "recently popular UQ methods for LLMs" and "most mainstream LLM UQ methods", are imprecise, as they do not specify that the paper discusses methods ultimately based on semantic clustering (even those papers cited in the last reply).

Examples of methods not covered by the position include Malinin & Gales (mentioned in the review) and all subsequent pure information-theoretic token-level approaches. This is why I suggested that the authors explicitly specify it, as the current position is too general for the arguments provided.

Regarding W2 and P(true), I feel there is an overload of the term "internal consistency". For pure semantic methods that require generating multiple samples, consistency was defined as "similarity between different generations" (sections 2.1 and 2.2). In section 2.3, what the authors called "consistency with internal beliefs" is different and refers to the likelihood that the generation is correct under the model.

But are these things equivalent? Does high P(true) correspond to the case where several generations from the model are semantically close? Maybe, but it is not clear, so I suggested that this example is not coherent.

Overall, I am still leaning toward rejection. I would rely on discussions with other reviewers and the AC for any changes to the assessment.

**Paper Summary:**

The paper introduces a position that "UQ in LLMs is mechanically isomorphic to an unsupervised clustering problem, which measures internal consistency and fails to serve as a practical safeguard".

The authors say that some popular UQ approaches in LLMs, specifically Semantic Entropy, Graph-based Quantification, and Verbalized uncertainty, are special cases of explicit/implicit clustering.

Authors discuss why the clustering of the model outputs is not the same as actual correctness. Specifically, they highlight that clustering is failing because UQ methods depend on sensitive parameters, clustering does not conflate confident hallucinations, and there is a lack of "true uncertainty".

Authors suggest improving the current situation by shifting to the "three-pillar paradigm": evaluation, mechanism, and grounding. For evaluation, the authors suggest analyzing worst-case robustness and reporting the sensitivity of the metrics to hyperparameters. For mechanisms, they suggest using methods that provide guarantees (e.g., conformal prediction) and utilizing post-training for uncertainty. For grounding, they suggest "gold-standard unit tests" and atomic fact verification.

**Position:**

Yes

**Position In Title:**

Yes

**Related Work:**

2

**Strengths And Weaknesses:**

## Strengths:

1. Overall, the paper is well structured and easy to read, clearly states the position, and suggests improvements in the current methodology.

## Weaknesses:

1. The position is not sufficiently argued.
- The paper considers only a specific class of the UQ methods, based on consistency. Specifically, semantic entropy (and related methods) and graph-based quantification require generating multiple sequences for a single prompt and then using an NLI model to evaluate their "similarity". However, these methods represent only a part of the methods used for UQ in LLMs. There is also a wide class of methods that work with token/sequence level probabilities and compute measures based on them (see, e.g., [1,2]). The paper under review does not consider them while saying "prevailing UQ" (line 015), "UQ in LLMs" (line 021, right), "current UQ for LLMs" (in the Conclusion), and, in the title itself, therefore overclaiming the generality of UQ measures. I suggest that authors specify the claims and be explicit that their position relates only to consistency-based UQ measures.
- In section 2, the authors conflate two different types of clusters. The first type (sections 2.1 and 2.2) is clustering based on semantic meanings, which requires generating different sequences. The measures of uncertainty based on this clustering assess the semantic consistency of the outputs. The second type of "clustering" is, informally speaking, a binary classification that distinguishes between two clusters -- p(true) and p(false). Even if one calls it clustering, I would argue that it is not about "internal consistency", as stated in position, line 023, right column. The generation of only **one** sequence says nothing about the internal consistency of the model. So, I would kindly disagree with the authors and would say that Subsection 2.3 is not coherent with the paper's narrative.
- More on the technical side: In 5.2, the authors suggest using conformal prediction to have "native guarantees". Authors suggest using the Efficiency (Set Size) of the conformal set as a "truth-aware" metric. Authors say, "In cases of 'confident hallucination,' this results in a Set Size Explosion, where the prediction set grows to encompass a massive portion of the vocabulary." First, it is not clear from the section what exactly is conformalized in the suggested setting. Second, in practice, only marginal conformal guarantees are feasible, and for a specific input, the set size itself can be misleading. Hence, the claim "set size is the only metric that honestly captures this failure mode" is too optimistic. I would like to know which exact combination of CP and LLMs the section refers to, and would be grateful if the authors could elaborate.

2. While the paper clearly states the position, it does not address it properly.
- First, the paper does not formally introduce what constitutes predictive uncertainty. I feel that, across the papers the authors refer to, different groups interpret "predictive uncertainty" differently. Hence, the right thing to start with is to clarify the author's point of view, what they imply by it. Is it pointwise model bias only (per input error)? Or is it the "variance" of the model's outputs (typically called consistency)? Or is it all merged?
- Second, the position is related to theoretical works on UQ. For example, [3] introduces epistemic uncertainty objectively as the divergence between a ground-truth reference and the model, including both model bias and variance (consistency). Hence, ground-truth TU can only be 0 when there is no mistake, and the model confidently predicts the correct label. On the contrary, in [4], the authors consider the EU subjectively (no bias is included) and focus only on the variance (consistency) of predictions. Hence, it is enough for TU to be 0 when the model collapses to the corner of a simplex (in the case of classification), regardless of whether it is correct or not.
I feel the authors imply that the first option (objective uncertainty, TU=0 when no mistake) is the desiderata, but they do not clarify their view.

----

References:

[1] - Malinin, A., & Gales, M. (2020). Uncertainty estimation in autoregressive structured prediction. arXiv preprint arXiv:2002.07650.

[2] - Vashurin, R., Goloburda, M., Ilina, A., Rubashevskii, A., Nakov, P., Shelmanov, A., & Panov, M. (2025). Uncertainty quantification for llms through minimum bayes risk: Bridging confidence and consistency. arXiv preprint arXiv:2502.04964.

[3] - Kotelevskii, N., Kondratyev, V., Takác, M., Moulines, É., & Panov, M. From risk to uncertainty: Generating predictive uncertainty measures via bayesian estimation, ICLR 2025.

[4] - Sale, Y., Bengs, V., Caprio, M., & Hüllermeier, E. (2024, July). Second-order uncertainty quantification: a distance-based approach. In Proceedings of the 41st International Conference on Machine Learning (pp. 43060-43076).

**Support:**

2

---

> ### Author Rebuttal · Authors · 2026-03-29
>
> We sincerely thank the reviewer for the constructive and detailed feedback. Below are our responses:
>
> > W1 & Q2: The paper only considers consistency-based UQ methods and overclaims generality. Does the position hold for token/sequence-level probability methods as well?
>
> Answer: Yes, our position mostly holds for these methods as well. Our paper already considers token/sequence-level probabilities. Semantic Entropy (SE), which we discuss extensively, is built on top of token/sequence probabilities. SE aggregates sequence-level probabilities into semantic clusters because raw token-level methods have well-documented limitations in LLM free-form generation. The evolution from raw token entropy to SE is itself evidence that the field has recognized the need for multi-sample semantic aggregation, which is the clustering paradigm we describe.
>
> Regarding the reviewer's references. [1] Malinin & Gales propose an ensemble-based framework for machine translation and speech recognition, requiring multiple independently trained model copies. This is nearly computationally infeasible at modern LLM scale. [2] Vashurin et al. actually reinforce our argument. This method combines token probability methods with semantic consistency computed over multiple sampled generations. Even methods starting from token probabilities end up requiring clustering to work effectively in the LLM setting.
>
> However, we do not claim every possible UQ method falls under the clustering framework. We observe that the mainstream has converged on this paradigm, and hope our diagnosis encourages exploration of fundamentally different directions. We will revise the manuscript to make the scope clearer.
>
>
> >W2: P(true) (Section 2.3) is not coherent with the paper's narrative. It is binary classification, not about "internal consistency," since only one sequence is generated.
>
> Answer: We agree that P(true) does not measure consistency between multiple generations. However, P(true) measures consistency with the model's internal beliefs. It checks whether a generated answer falls within the high-confidence region of the model's latent space.
>
> We acknowledge this is mechanistically different from SE's semantic clustering or graph-based spectral clustering. Section 2.3 is not meant to claim the same mechanism as semantic clustering, but rather to highlight a shared limitation: all three families remain self-referential and rely on internal model signals rather than external truth. We will revise the text to make this distinction explicit and clarify that the claim of Section 2 is about dependence on internal model states, not identical clustering mechanics.
>
>
> >W3: The claim "set size is the only metric that honestly captures this failure mode" is too optimistic
>
> Answer: To clarify, our paper does not propose CP as a new UQ method. We propose that CP could serve as an evaluation framework for assessing existing UQ methods. If a UQ method's confidence scores are used as nonconformity scores in CP, the resulting prediction set size reveals the informativeness of those scores. Our claim about set size refers specifically to confident hallucination within this evaluation framework. We also welcome other evaluation metrics as shown in Section 5.1.1. We acknowledge that marginal coverage does not imply conditional guarantees and will revise the language to be more precise.
>
> >W4: "Is it pointwise model bias only? Or is it the 'variance' of the model's outputs? Or is it all merged?"
>
> Answer: The term "predictive uncertainty" does not appear in our manuscript. Our paper addresses overall predictive uncertainty without decomposing it into bias and variance, consistent with the majority of UQ works in the LLM domain [1]. We will add a formal statement in the revised introduction
>
> [1] Lambert, Benjamin, et al. "Trustworthy clinical AI solutions: a unified review of uncertainty quantification in deep learning models for medical image analysis." Artificial Intelligence in Medicine.
>
> Q1: Do authors mean to define it objectively (UQ = 0 iff no mistake)?
>
> Answer: Our position aligns with the objective definition in the reviewer's [3], where TU = 0 only when the model confidently predicts the correct answer, rather than the subjective definition in [4]. We believe this objective notion is also the implicit assumption of most existing LLM UQ works, since dominant evaluation metrics (AUROC, AUPRC) are defined by the correlation between uncertainty and answer correctness. A method scores well on AUROC when high uncertainty corresponds to incorrect answers, which presupposes TU should not be 0 when the model is confidently wrong
>
> In Section 4, Objection 2, we acknowledge that measuring subjective belief has research value. However, our position is that practical utility for trustworthy deployment demands an objective notion. We will formalize this in the paper, and we are grateful for these theoretical pointers
>
> **`We'd be happy to further discuss should any concern remains!`**

---

> > ### Author Rebuttal · Reviewer_oBYr · 2026-04-03
> >
> > I want to thank the authors for their reply. Please, find my comments below.
> >
> > W1. I agree with the authors' view on [1]. However, this aligns with my original concern. The paper [1] represents another family of UQ methods for LLMs that is not covered in this position. That's why I believe it is more appropriate for authors to frame the scope of the methods considered (semantic methods).
> > Regarding SE, it indeed uses sequence probabilities, with NLI applied on top. However, in [1], it is used differently.
> >
> >
> > W2. Thank you for your answer. I still think that p(true) is not a coherent example here.
> >
> > Regarding the authors' reply, I have two remarks. I did not quite understand what is meant by "consistency with the model's internal beliefs".
> >
> > Also, authors say, "all three families remain self-referential and rely on internal model signals rather than external truth." I agree, but this applies broadly to any method that does not use external information, and is not specific to the LLM/clustering setting discussed here.
> >
> > W3. Thank you for the reply. I understand that the paper does not provide a new UQ method based on CP. But it remains unclear how, given CP's well-known limitations (e.g., marginal guarantees, calibration data requirements, and failure modes with set sizes), the authors expect to evaluate LLMs with CP.
> >
> > W4 and Q1. Thank you for your answers. This addresses my concerns well.
> >
> > I would only remark that, since the authors treat TU = 0 as meaning no mistake is possible, they take both model bias and predictive variance into account.
> >
> > Regarding the term "predictive uncertainty", I believe it better describes uncertainty with respect to what we consider. There also could be "input ambiguity/uncertainty", but this is not the point of the paper.

---

### Official Review · Reviewer_vckF · 2026-03-12

**Significance:** 3
**Argument Clarity:** 3
**Rating:** 4
**Confidence:** 4

**Questions:**

For the rebuttal, please justify the inconsistency in reasoning mentioned in the weakness section

**Alternative Views Section:**

Yes

**Compliance With Llm Reviewing Policy A Conservative:**

Affirmed.

**Discussion Potential:**

4

**Final Justification:**

I keep my initial score as my concerns are addressed.

**Paper Summary:**

The paper presents the position that UQ in LLMs is toward the pursuit of unsupervised clustering rather than factual inaccuracy. It is discussed how popular UQ methods (semantic clustering, graph-based, and p(True) demonstrate the clustering mechanism. Next, the paper discusses the failure modes of clustering in UQ: parameter sensitivity, confident hallucination, and lack of ground truth. Additionally, the paper discusses three alternative views and the subsequent avenues for research.

**Position:**

Yes

**Position In Title:**

Yes

**Related Work:**

3

**Strengths And Weaknesses:**

Strengths
- With state-of-the-art UQ methods such as semantic entropy, kernel language entropy, and spectral clustering in the literature, the paper offers a timely alternative perspective on the direction of UQ in open-ended generation settings.
- The paper is well motivated and written: (i) First, it demonstrates how the popular methods come under the section of clustering, (ii) Second, it discusses the critical failure scenarios of the clustering-based methods, and (iii) Finally, it discusses the potential avenues to address the critical failure scenarios.


Weakness
- One major criticism of the paper is that, though most of the methods are portrayed as clustering, the failure scenarios of clustering are argued for methods that rely on multiple answer generation. In parameter sensitivity (value of n, NLI) and confident hallucination (self-consistency), the reasoning does not necessarily apply to UQ that rely on single sample generation:  perplexity, [1], [2], [3]. This inconsistency in reasoning weakens the evidence of the position.

References:

[1] Kadavath, Saurav, et al. "Language models (mostly) know what they know." arXiv preprint arXiv:2207.05221 (2022).

[2] Azaria, Amos, and Tom Mitchell. "The internal state of an LLM knows when it’s lying." Findings of the Association for Computational Linguistics: EMNLP 2023. 2023.

[3] Bakman, Yavuz Faruk, et al. "MARS: Meaning-aware response scoring for uncertainty estimation in generative LLMs." Proceedings of the 62nd Annual Meeting of the Association for Computational Linguistics (Volume 1: Long Papers). 2024.

**Support:**

3

---

> ### Author Rebuttal · Authors · 2026-03-29
>
> We sincerely thank the reviewer for the positive assessment and for recognizing our paper's timely alternative perspective. Here is our response:
>
> >W1: The failure scenarios (parameter sensitivity, confident hallucination) are argued for multi-sample methods, but do not necessarily apply to single-sample UQ: perplexity, [1] Kadavath et al. (P(true)), [2] Azaria & Mitchell, [3] MARS.
>
> Answer: We appreciate this observation and would like to address it from two angles.
>
> First, regarding the landscape of LLM UQ methods. Pure single-sample methods like perplexity have been shown to be insufficient for LLM free-form generation, which is exactly why the community has moved toward multi-sampling approaches. The current mainstream, including Semantic Entropy, graph-based methods, and their variants, all rely on generating multiple responses and performing some form of aggregation. Even MARS [3], while proposing a better token-level scoring function, is designed to be integrated with multi-sample methods (e.g., SE+MARS in their experiments) and its own evaluation relies on multiple sampled generations. Our paper focuses on this mainstream precisely because these are the methods being actively developed and the direction the field has converged on.
> Second, regarding the specific single-sample methods the reviewer mentions. P(true) [1] we discuss extensively in Section 2.3, where we argue it measures consistency with internal beliefs rather than factual correctness. Azaria & Mitchell [2] trains a classifier on the model's internal hidden states to predict correctness. This is a supervised approach that uses labeled data for training, so it already moves in the direction we advocate in Section 5 and is closer to our proposed solution than to the unsupervised methods we critique.
>
> That said, we do not claim that every possible UQ method must fall under our clustering framework. As with our response to Reviewer oBYr, we observe that the current mainstream has converged on this paradigm, and we hope our framework can help the community recognize this structural pattern and explore fundamentally different directions. We will revise the manuscript to make the scope of our discussion more explicit.
>
> [1] Kadavath, Saurav, et al. "Language models (mostly) know what they know." arXiv preprint arXiv:2207.05221 (2022).
>
> [2] Azaria, Amos, and Tom Mitchell. "The internal state of an LLM knows when it’s lying." Findings of the Association for Computational Linguistics: EMNLP 2023. 2023.
>
> [3] Bakman, Yavuz Faruk, et al. "MARS: Meaning-aware response scoring for uncertainty estimation in generative LLMs." Proceedings of the 62nd Annual Meeting of the Association for Computational Linguistics (Volume 1: Long Papers). 2024.

---

> > ### Author Rebuttal · Reviewer_vckF · 2026-04-02
> >
> > Thank you for clearing my concerns.
> >
> > My concerns are addressed, but the inconsistencies while discussing weakness of current methods should be revised in the manuscript, if the paper is accepted.
> > I stick to my current positive score.

---

### Official Review · Reviewer_oJyG · 2026-03-16

**Significance:** 3
**Argument Clarity:** 4
**Rating:** 5
**Confidence:** 5

**Questions:**

To what extent do you think the arguments in this paper apply to UQ methods in general and not just the bespoke UQ methods for LLMs. Can UQ methods used for standard image classification tasks, such as Variational Training, Laplace Approximations, Deep Ensembles, Conformal Predictions etc, also be recast as unsupervised clustering and do they also suffer from the same issues outlined above? If not, do the problems with UQ in LLMs stem from the specific choice of methods, or are they more fundamental problems with how we conceive of UQ in the first place?

**Alternative Views Section:**

Yes

**Compliance With Llm Reviewing Policy A Conservative:**

Affirmed.

**Discussion Potential:**

3

**Paper Summary:**

This paper critically analyzes standard uncertainty quantification metrics for LLMs. They survey the major classes of uncertainty quantification metrics and make an argument for why they all approximately correspond to unsupervised clustering algorithms. They note three failure modes that arise from methods that follow this structure: 1) hyperparameter sensitivity 2) internal evaluation
cycle 3) lack of ground truth.

**Position:**

Yes

**Position In Title:**

Yes

**Related Work:**

3

**Strengths And Weaknesses:**

# Strengths
1. The authors did a great job of extensively outlining the alternative views, representing the arguments more or less fairly, and provided a cogent response to each.
2. I also appreciate that while the main focus of the paper was critical analysis of a certain class of methods, the author also proided constructive methodological suggestions that might resolve some of the issues outlined in the paper. I think the suggestions outlined in section 5 make a lot of sense and constitute a viable direction for future research.

# Weakness
I think the argument is a bit loose in some places. Specifically, how accurate is casting Internal Verbalized Beliefs as clustering (section 2.3). It's not obvious to me that this is accurate. More importantly, I am not sure I see the importance of recasting uncertainty quantification methods as clustering, which is bit contrived. As far as I see the chain of argument is 1) Uncertainty Quantification is Unsupervised clustering 2) Unsupervised clustering suffers from problem X, thus Uncertainty Quantification suffers from problem X. However, of the problems outlined in section 3, only 3.1 (Hyperparameter sensitivity) directly refers back to clustering, and even this argument could have been made without reference to clustering. So I am not sure why this step was necessary at all.

**Support:**

4

---

> ### Author Rebuttal · Authors · 2026-03-29
>
> We thank the reviewer for the positive assessment and for finding our alternative views section fair and constructive suggestions viable. Here are our responses:
>
> > **W1: Casting P(true) as clustering is a bit loose. More importantly, why is the clustering framing necessary at all? The problems in Section 3 could be argued without reference to clustering.**
>
> **Answer:**
> **On P(true) as clustering.** We use "clustering" for P(true) to refer to partitioning the model's latent space into confidence regions, which represents consistency with internal beliefs rather than consistency between generations. When P(true) evaluates a generated answer, it checks whether that answer is consistent with what the model internally considers to be "true," not whether the answer is factually correct. Figure 2 empirically supports this. High P(true) and low P(true) samples form geometrically separated clusters in hidden state space, showing that P(true) is partitioning the latent space into confidence regions. We acknowledge this is mechanistically different from the explicit semantic clustering in SE or spectral clustering in graph-based methods, and we will clarify this distinction in the revision.
>
> **Why we need the clustering framework.** The reason we chose this framing is that we observed a strong commonality across the major UQ methods. Despite their surface-level diversity, they all converge on the same unsupervised paradigm of measuring internal agreement. Unsupervised clustering, as a field, has well-known and arguably fundamental limitations. It can tell you how well-separated your groups are, but it cannot tell you whether those groups correspond to anything meaningful in the real world without external labels. We believe the UQ community has inherited exactly this limitation, and many researchers may not be fully aware of it because each method appears distinct on the surface.
>
> By explicitly naming this shared structure as "clustering," we hope to achieve two things. First, we want to warn the community that incremental improvements within this paradigm (better NLI models, better graph structures, better scoring functions) cannot resolve the fundamental problem, because the problem is structural rather than technical. Second, and more importantly, we hope this framing can help researchers jump out of this paradigm entirely and explore fundamentally different directions, such as the externally grounded approaches we propose in Section 5.
>
>
> > **Q1: Do these arguments apply to UQ beyond LLMs (e.g., Deep Ensembles, Laplace Approximations for image classification)? Do problems stem from method choices or from how we conceive of UQ?**
>
> **Answer:** Excellent question. We argue the problems are **largely specific to the LLM setting**, which is why our title scopes the claim to UQ in LLMs.
>
> Traditional UQ methods for image classification operate in a closed-label space where ground truth is well-defined, and calibration can be directly measured. Deep Ensembles capture epistemic uncertainty through disagreement across independently trained models. This is fundamentally different from sampling multiple outputs from a single LLM, which only captures variation within one model's learned distribution. The clustering pathology manifests acutely in LLMs for three reasons. First, the output space is open-ended, making ground truth harder to define. Second, LLMs can be confidently wrong due to training biases and RLHF-induced overconfidence. Third, the scale of modern LLMs makes true ensemble approaches infeasible, forcing the community toward single-model heuristics operating on internal signals alone.

---

> > ### Author Rebuttal · Reviewer_oJyG · 2026-04-02
> >
> > I am happy with the response. I will keep my positive score.

---

### Decision · Program_Chairs · 2026-04-30

**Decision:**

Accept (regular)

**Comment:**

Overall, the majority of reviewers find the clustering interpretation proposed here to provide an interesting view of weaknesses of many existing UQ methods for LLMs.  There were some detailed concerns which were mostly addressed by the author rebuttals; please take care to incorporate these into the manuscript, especially the clarifications about "P(True)" methods which caused the most confusion.  Please also revise to better indicate the scope of the paper (range of UQ methods covered) in the introduction, and clarify how these differ from some other references highlighted by reviewer oBYr.